# Fungal Immunomodulatory Protein from *Nectria haematococca* Suppresses Growth of Human Lung Adenocarcinoma by Inhibiting the PI3K/Akt Pathway

**DOI:** 10.3390/ijms19113429

**Published:** 2018-11-01

**Authors:** Yingying Xie, Shuying Li, Lei Sun, Shujun Liu, Fengzhong Wang, Boting Wen, Lichao Sun, Xiangdong Fang, Yushuang Chai, Hao Cao, Ning Jia, Tianyi Gu, Xiaomin Lou, Fengjiao Xin

**Affiliations:** 1Laboratory of Food Enzyme Engineering, Institute of Food Science and Technology, Chinese Academy of Agricultural Sciences, Beijing 100193, China; 15210650462@163.com (Y.X.); lishuying@caas.cn (S.Li.); LolyLiu0224@163.com (S.Liu.); wangfengzhong@caas.cn (F.W.); wenboting@caas.cn (B.W.); sun2004go@163.com (L.S.); caohao@mail.buct.edu.cn (H.C.); jianing0805@126.com (N.J.); 18501151081@163.com (T.G.); 2CAS Key Laboratory of Genome Sciences and Information, Beijing Institute of Genomics, Chinese Academy of Sciences, Beijing 100101, China; sunleixy@126.com (L.S.); fangxd@big.ac.cn (X.F.); 3University of Chinese Academy of Sciences, Beijing 100049, China; 4Sino-Danish College, University of Chinese Academy of Sciences, Beijing 101408, China; 5Guangzhou Baiyunshan Zhongyi Pharmaceutical Co., Ltd., Guangzhou 510530, China; chaiys09@163.com

**Keywords:** fungal immunomodulatory protein, *Nectria haematococca*, lung adenocarcinoma, PI3K/Akt, apoptosis, autophagy, cell cycle arrest

## Abstract

Lung cancer is a common disease that is associated with poor prognosis. Fungal immunomodulatory protein from *Nectria haematococca* (FIP-nha) has potential as a lung cancer therapeutic; as such, illuminating its anti-tumor mechanism is expected to facilitate novel treatment options. Here, we showed that FIP-nha affects lung adenocarcinoma growth ex vivo and in vivo. Comparative quantitative proteomics showed that FIP-nha negatively regulates PI3K/Akt signaling and induces cell cycle arrest, autophagy, and apoptosis. We further demonstrated that FIP-nha suppresses Akt phosphorylation, leading to upregulation of p21 and p27 and downregulation of cyclin B1, cyclin D1, CDK2, and CDK4 expression, ultimately resulting in G1/S and G2/M cell cycle arrest. Meanwhile, FIP-nha-induced PI3K/Akt downregulation promotes A549 apoptosis by increasing the expression ratio of Bax/Bcl-2 and c-PARP and autophagy by decreasing the phosphorylation of mTOR. Thus, we comprehensively revealed the anti-tumor mechanism of FIP-nha, which inhibits tumor growth by modulating PI3K/Akt-regulated cell cycle arrest, autophagy, and apoptosis, and provided the basis for further application of fungal immunomodulatory proteins, especially FIP-nha.

## 1. Introduction

Lung cancer is the first and second leading cause of cancer-related mortality among men and women worldwide, respectively [1]. Lung adenocarcinoma, a form of non-small cell lung cancer (NSCLC), is the most common histologic type of lung cancer, and accounts for 30–35% of all primary lung tumors [2]. Although significant achievements have been made in the treatment of lung cancer, the prognosis for patients with advanced NSCLC still remains poor [3]. Considering the poor efficacies and severe toxicities associated with existing chemotherapeutics in the treatment of lung cancer, there has been increased demand for the development of new drugs and functional foods for this application, especially those derived from natural compounds [4].

Natural products are an effective source of anti-tumor agents [5,6]. It is estimated that more than 80% of anti-cancer drugs are either natural products per se or are based thereon [7]. Fungal products display widespread biological activities such as anti-inflammation, anti-cancer, and immunomodulatory effects, and thus these species represent a wealth of functional foods and drug precursors [8,9]. Mounting evidence suggests that fungal extracts might protect against certain types of cancers, especially lung adenocarcinoma [10,11]. Among them, fungal immunomodulatory proteins (FIPs), which are widely found in fungi, have well-known, strong inhibitory effects on lung adenocarcinoma. FIPs comprise a novel and important family of bioactive proteins, and to date more than 10 of these have been isolated and identified, such as FIP from *Ganoderma lucidum* (LZ-8 or FIP-glu) [12], *Ganoderma tsugae* (FIP-gts) [13], *Flammulina velutipes* (FIP-fve) [14], and *Nectria haematococca* (FIP-nha) [15]. Known FIPs exhibit high similarities in terms of amino acid sequences (>50%) and protein structure [16].

Despite high sequence identity, the anti-cancer mechanisms of different FIPs are not identical. In lung adenocarcinoma cells, FIP from *Ganoderma microsporum* (FIP-gmi) activates autophagy to inhibit cancer cell growth [17]; furthermore, FIP-gts induces premature senescence [18], whereas FIP-SN15 [19], FIP-glu [20], and FIP-sch2 promote apoptosis [16]. This discrepancy could be partly due to the different properties of FIPs as well as incomplete studies that have not fully demonstrated associated anti-tumor mechanisms.

FIP-nha from *Nectria haematococca* consists of 114 amino acid residues, and its molecular weight is 12,837 Da. FIP-nha has been revealed to have potent anti-tumor effects against HL60, HepG2, and MGC823 cells via the induction of apoptosis [15]. Our previous study showed that FIP-nha greatly suppresses the growth of A549 cells, and its efficacy was found to be superior to that of FIP-fve and LZ-8, which indicates that it has promise for medical applications. A thorough understanding of the associated mechanisms and proper design of therapeutic approaches are critical for effective cancer therapeutics. Accordingly, the molecular mechanism and effects of FIP-nha on lung adenocarcinoma are urgently needed to be illuminated.

Proteomic analysis can indicate overall alterations in protein expression levels, and this knowledge contributes to a comprehensive understanding of the molecular response to stimuli. Therefore, in the present study, we performed comparative quantitative proteomics to identify differential protein expression induced by FIP-nha in A549 cells. Combined with a series of convincing results acquired by flow cytometry and western blotting analysis, we aimed to provide new knowledge regarding the mechanism through which FIP-nha inhibits lung adenocarcinoma growth.

## 2. Results

### 2.1. FIP-nha Inhibits Lung Adenocarcinoma Cell Growth Ex Vivo and In Vivo

The effects of FIP-nha on A549 and NCI-H2347 (H2347) human adenocarcinoma and MRC-5 human lung fibroblast cell growth were investigated by performing MTT assays. FIP-nha significantly inhibited A549 and H2347 cell growth, and the inhibition was dose-dependent. Moreover, no prominent effect on MRC-5 cell growth was observed (Figure 1A). These results suggested that FIP-nha selectively inhibited lung adenocarcinoma cells, but not normal cells.

To further evaluate its anti-tumor activity in vivo, FIP-nha was injected intraperitoneally into nude mice subcutaneously inoculated with A549 cells. The average tumor volume in the high-dose FIP-nha treatment group (40 mg/kg mouse body weight, N = 8) was significantly decreased compared to that in the negative control group at day 33, and the efficacy of this treatment was equivalent to that of the chemotherapy drug doxorubicin (4 mg/kg mouse body weight, N = 8). Low-dose FIP-nha treatment (20 mg/kg mouse body weight) had a weak but insignificant inhibitory effect (Figure 1B,C). Mouse body weights in FIP-nha-treated and untreated groups were further compared, and FIP-nha treatment had no effect on the average mouse body weight (Figure 1D). Taken together, FIP-nha inhibited lung adenocarcinoma cell growth ex vivo and in vivo.

### 2.2. Proteomic Analysis of FIP-Nha-Induced Differential Expressed Proteins in A549 Cells

Aiming to overcome the lack of effective information regarding mechanisms associated with the effects of FIP-nha, isobaric tags for relative and absolute quantification (iTRAQ)-based quantitative proteomic analysis was applied to globally profile protein alterations in A549 cells and to explore the anti-tumor mechanism associated with this compound. The flow diagram of the proteomic analysis is shown in Figure 2A; the peptides from one biological replicate of PBS- and FIP-nha-treated cells were labeled with 119 and 113 tags, and those from the other biological replicates were labeled with 121 and 114 tags, respectively. Finally, 18,356 unique peptides corresponding to 2650 proteins were identified, and 99% of these were identified based on iTRAQ tags. FIP-nha-regulated proteins were defined using the following criteria: (1) *p* ≤ 0.05; (2) fold change ≥ 1.2 or ≤ 0.83; (3) proteins in biological duplicates had the same change trends (Figure 2B). As a result, 334 differentially-expressed proteins were identified, including 213 up- and 121 downregulated proteins (Figure 2C).

To provide additional information regarding the mechanism associated with FIP-nha, we further classified the up- and downregulated proteins into different functional categories. This analysis indicated that the upregulated proteins were mainly involved in extracellular matrix organization, PI3K/Akt signaling, cell apoptosis, cell autophagy, and cell migration (Figure 2D), and downregulated proteins were enriched in functions including G1/S and G2/M cell cycle arrest, ubiquitination, telomere maintenance, and cell proliferation (Figure 2E). Dysregulated PI3K/Akt signaling has been implicated in many cancers and is directly related to cellular quiescence, proliferation, apoptosis, and autophagy, indicating that FIP-nha likely activates apoptosis, autophagy, and cell cycle arrest by regulating the PI3K/Akt signaling pathway.

### 2.3. FIP-Nha Negatively Regulates the PI3K/Akt Signaling Pathway and Induces Cell Autophagy

Given the results of functional category enrichment analysis and the critical role of the PI3K/Akt pathway in lung adenocarcinoma cells, the involvement of this pathway in FIP-nha-mediated cell growth inhibition was investigated. As shown in Figure 3A, in both A549 and H2347 cells treated with different concentrations of FIP-nha (0, 4, 8, and 16 µg/mL) for 24 h, the levels of phosphorylated Akt (p-Akt) at Thr^308^ along with its downstream effector phosphorylated mammalian target of rapamycin (p-mTOR) were significantly suppressed in a dose-dependent manner compared to those in the untreated group.

PI3K/Akt/mTOR signaling plays a crucial role in regulating autophagy, therefore, an autophagy marker, namely microtubule-associated protein light chain B (LC3B) protein, was used to assess whether FIP-nha suppressed autophagy in lung adenocarcinoma cells. Figure 3A shows that A549 and H2347 cells treated with FIP-nha expressed increasingly high levels of LC3B-II in a dose-dependent manner, indicating that autophagy might be activated by FIP-nha. Ultrastructural analysis by transmission electron microscope (TEM) revealed FIP-nha treatment resulted in the formation of numerous autophagic vacuoles in A549 cells (Figure 2B). Furthermore, bafilomycin A1 which is a vacuolar-type H^+^-ATPase inhibitor and blocks autophagosome-lysosome fusion was used to analyze autophagic flux. As shown in Figure 3C, treatment with FIP-nha combined bafilomycin A1 induced more LC3B-II accumulation. These results suggested that FIP-nha induced autophagy of human lung adenocarcinoma cell.

### 2.4. FIP-Nha Induces G1/S and G2/M Cell Cycle Arrest in Lung Adenocarcinoma Cells

To determine if FIP-nha suppresses cell growth through the establishment of a cell cycle checkpoint, A549 and H2347 cells were cultured in the absence or presence of 8 µg/mL FIP-nha, stained with propidium iodide (PI), and analyzed by flow cytometry. An enhanced population of G1 cells was observed in the treated group compared to that in the control group in both A549 and H2347 cells (Figure 4A), indicating that FIP-nha treatment results in G1/S cell cycle arrest, but not G2/M arrest. However, proteomic analysis suggested that FIP-nha might induce both G1/S and G2/M cell cycle arrest; we hypothesized that progression through G1/S was blocked in the majority of cells, resulting in a relative reduction of cells in the G2 phase.

To verify this hypothesis, cell cycle regulatory proteins including cyclin B1, cyclin D1, CDK2, and CDK4 were assessed by western blotting. The levels of these proteins were all downregulated in FIP-nha-treated cells compared to those in control cells (Figure 4B). In addition, expression levels of inhibitory cell cycle regulators, p21 and p27, were upregulated with FIP-nha treatment (Figure 4B). Therefore, FIP-nha inhibited lung adenocarcinoma cell growth by blocking G1/S and G2/M phase transitions.

### 2.5. FIP-Nha Promotes Apoptosis in Lung Adenocarcinoma Cells

Proteomic analysis showed that FIP-nha treatment promoted apoptosis in A549 cells. To verify this result, Annexin V-FITC and PI staining were applied. As shown in Figure 5A, apoptotic rates of A549 and H2347 cells were increased to 52.95% and 59.11% from 5.70% and 10.56% respectively when the cells were treated with FIP-nha (8 µg/mL) for 24 h. To further investigate the underlying mechanisms related to FIP-nha-induced apoptosis, the levels of several key apoptosis-related proteins were determined by western blotting. In both A549 and H2347 cells, the apoptosis-related proteins c-PARP and Bax were upregulated following FIP-nha treatment, whereas Bcl-2, an anti-apoptotic marker, decreased (Figure 5B). These data indicated that FIP-induced apoptosis might also contribute to the inhibition of lung adenocarcinoma cell growth. To elucidate the relationship between autophagy and apoptosis in our model, A549 cells were first treated with an inhibitor of autophagy, namely 3-methyladenine, and then FIP-nha; here, inhibition of autophagy was found to increase the apoptotic response (Appendix A). Therefore, FIP-nha-induced autophagy might facilitate the survival of tumor cells rather than induce cell death.

### 2.6. FIP-Nha Affects Cell Cycle, Autophagy, and Apoptosis Processes in Xenograft Mouse Tumors

The changes of cell cycle-, autophagy-, and apoptosis-related protein expressions were further confirmed in xenograft tumor of A549 cells. After treatment with FIP-nha (40 mg/kg mouse body weight), the levels of p-Akt, cyclin B1, cyclin D1 and Bcl-2 decreased, and expressions of c-PARP, Bax and LC3B-II increased, indicating a good agreement with the results of the experiment in vitro (Figure 6A). In addition, the effect of FIP-nha on cell proliferation and apoptosis in vivo was assessed by immunochemistry (IHC) staining of cleaved caspase 3 and Ki-67. As shown in Figure 6B, FIP-nha treatment resulted in enhancement of cleaved caspase 3 expression and inhibition of Ki-67 expression in xenograft tumors of A549 cells. Collectively, these results suggested that FIP-nha inhibited lung adenocarcinoma cell growth through regulating cell cycle arrest, autophagy, and apoptosis ex vivo and in vivo.

## 3. Discussion

FIP-nha is from the pathogenic fungus, *Nectria haematococca*, whereas most other FIPs have been isolated and identified from edible mushrooms, such as LZ-8 from *Ganoderma lucidium*, FIP-fve from *Flammulina velutipes*, and FIP-gts from *Ganoderma tsugae*. Higher fungi are well-known for their high nutritional and medicinal value. Importantly, some mushrooms, as supplements in commercial anti-cancer drugs, have been found to work in synergy to treat drug-resistant cancers [21]. Findings have shown that the active ingredients with anti-cancer potential in mushrooms include lentinan, krestin, ganoderic acid, lectin, psilocybin, laccase, and schizophyllan, among others. Among these, polysaccharides, as potential anti-tumor agents, are the best known; nevertheless, the most diverse fungal-derived proteins that exhibit strong anti-tumor and immunomodulatory properties are less studied. Understanding the bioactive proteins in foods will help to uncover a wealth of promising anti-tumor drugs, will provide better insight into the mechanisms underlying their biological action, and will promote their pharmaceutical application for cancer therapy. Therefore, our study not only provides the anti-tumor mechanism of one fungus-derived protein, namely FIP-nha, but also will encourage more discovery- and mechanism-related research on other natural microbial proteins.

Constitutive activation of PI3K/Akt signaling contributes to the progression of lung cancer [22,23]. The PI3K/Akt pathway controls multiple biological processes [24,25]. First, to promote cell survival, PI3K/Akt signaling regulates the levels of cyclin-dependent kinase inhibitors p21 and p27 and thus cell cycle [26]; specifically, p21 is involved in arresting cells in both G1 and G2 cell cycle phases, whereas p27 blocks G1/S-phase transition [27]. FIP-nha increased p21 and p27 levels through the negative regulation of PI3K/Akt signaling to induce growth arrest, which further resulted in the downregulation of cyclin and CDK proteins (Figure 4B). Bcl-2 and Bax are the two principal members of the Bcl-2 multigene family. Bcl-2 prevents apoptosis, whereas Bax exhibits a pro-apoptotic effect, both of which are regulated by Akt [28,29]. Downregulation of Akt phosphorylation was found to increase the expression of Bax and decrease Bcl-2 levels, resulting in a significant increase in lung adenocarcinoma cell apoptosis (Figure 5B). mTOR is a downstream component of the PI3K/Akt signaling pathway, as well as a major regulator of apoptosis and the autophagic process [30,31]. The PI3K/Akt/mTOR pathway was found to be inhibited with FIP-nha-activated autophagy (Figure 3A). Collectively, our study revealed multiple FIP-nha-mediated proteins and their biological functions, thus contributing to studies regarding the efficient application of FIPs to lung cancer therapy.

Although FIPs are highly similar based on amino acid sequence and structure, their anti-cancer mechanisms have been found to be diverse. For example, in lung cancer cells, FIP-gts decreases telomerase expression through nuclear export mechanisms mediated by endoplasmic reticulum stress-induced intracellular calcium levels [32]. Despite the up to 66% identity between FIP-nha and FIP-gts, FIP-nha was not found to affect cellular senescence. In addition, inhibition of autophagy induced by FIP-nha leaded to enhancement of the apoptotic response, suggesting that FIP-nha-induced autophagy might protect the cells from injury, however, some FIPs, such as FIP-gmi, induced cell death through activating autophagy in A549 cells [17]. Both apoptosis and autophagy are cellular degradation pathways that are essential for organismal homeostasis [33]. Apoptosis is a process comprising the genetically-determined elimination of cells [34], whereas autophagy tends to recycle nutrients and balance sources of energy by clearing damaged proteins, organelles, pathogens, or aggregates [35]. Autophagy has a dual function: survival-promoting [36,37] and leading to the activation of apoptosis or cell death through a cell-autodigestive process [38,39]. Although both FIP-nha and FIP-gmi can activate autophagy of A549 cells, the ultimate effects as well as the underlying mechanisms may be different and deserve further investigation. These results also reflected that high similarity in protein sequence or structure is not equivalent to similar functions, which should be considered before their applications.

Comparative proteomics and functional enrichment analysis were performed to assess overall alterations at the protein level. In addition to the biological processes and pathways that were confirmed in our study, others might also be meaningful. Among them, FIP-nha induction promoted protein ubiquitination (Figure 2D,E). The relationship between ubiquitination and autophagy has been widely reported; the binding of autophagy receptors to both ubiquitin and LC3 controls protein degradation through the selective autophagy pathway [40]. In addition, changes to cell migratory abilities were also suggested by proteomic analysis, and this result was confirmed in A549 cells that FIP-nha significantly inhibited tumor cell migration. Similar studies have also been reported; for example, FIP-fve inhibits lung cancer cell migration via Rac GTPase activating protein 1 [41]. Therefore, our study only presented part of the anti-tumor functions and mechanisms of FIP-nha, and thus more of the proteomics data should be exploited in further studies.

Drug and food intake patterns are of great importance for therapeutic applications. Oral administration, the most common route, is convenient and cheap, and depends on stability against acidic and alkaline conditions and enzymes. Therefore, whether FIPs can maintain activity in the gastrointestinal tract is worth considering. Researchers have tested the stabilities of FIP-gts and FIP-fve, and have indicated that FIPs are highly resistant to acid hydrolysis, alkali decomposition, and enzyme digestion; they also possess the high potential for development as food or pharmaceutical products [42]. In our study, intraperitoneal injection—instead of oral administration—was used for FIP-nha treatment with a xenograft model to avoid the confounding effects of absorption efficiency and food interaction. However, the stability of FIP-nha should also be confirmed, despite the fact that it is highly similar to other stable FIPs based on amino acid sequences.

Collectively, our study comprehensively investigated the anti-cancer mechanism of FIP-nha using a proteomic approach and found that FIP-nha negatively regulated the PI3K/Akt signaling pathway, consequently inducing G1/S and G2/M cell cycle arrest by upregulating p21 and p27 and downregulating cyclin B1, cyclin D1, CDK2, and CDK4. This was further found to promote autophagy and apoptosis by decreasing mTOR and increasing Bax/Bcl-2 and c-PARP expression, respectively. We believe that FIP-nha, as well as other FIPs, might be promising therapeutic agents for the treatment of lung cancer.

## 4. Materials and Methods

### 4.1. Materials and Chemicals

The gene encoding FIP-nha (GenBank ID: EEU37941.1) was synthesized by Qinglan Biotech Co. (Wuxi, China). The plasmid pGEX-6P-1 and *Escherichia coli* competent cells used for protein expression were purchased from Amersham Pharmacia Biotech (Buckinghamshire, UK) and TransGen Biotech (Beijing, China), respectively. A549 and MRC-5 cells were obtained from the Cell Bank of Shanghai Institutes for Biological Sciences, Chinese Academy of Sciences (Shanghai, China). H2347 cells were kindly supplied by Ting Xiao from Chinese Academy of Medical Sciences and Peking Union Medical College. The primary antibodies were obtained from Santa Cruz Biotechnology (Santa Cruz Biotechnology, Santa Cruz, CA, USA; anti-β-actin, anti-GAPDH, anti-p-Akt, anti-p-mTOR, anti-p27, anti-cyclin D1, and anti-CDK4, used at 1:1000), Beyotime Biotechnology (Shanghai, China, anti-p21, anti-cyclin B1, anti-CDK2, anti-c-PARP, anti-Bcl-2, anti-Bax, anti-cleaved caspase 3 and anti-Ki-67, used at 1:1000 for western blotting and 1:200 for IHC), and Novus Biologicals (Littleton, CO, USA; anti-LC3B, used at 1:1000).

### 4.2. FIP-Nha Protein Expression and Purification

The gene encoding FIP-nha was cloned into the pGEX-6P-1 expression vector using BamH I and Xho I restriction sites, and the sequence encoding a GST tag was at the N-terminus of the gene. The recombinant plasmid was transformed into *E. coli* Rosetta (DE3) and cultured in LB media containing 1% ampicillin at 37 °C. When the OD_600_ reached 0.6, 0.1 mM isopropyl-β-d-thiogalactopyranoside was added to induce the expression of FIP-nha. After 16-h culture at 16 °C, total cells were collected and disrupted using a high-pressure homogenizer (ATS Engineering Inc., Dresden, Germany). The supernatant obtained by centrifugation was purified by affinity chromatography using a glutathione Sepharose column (Pharmacia, Stockholm, Sweden). The fusion proteins were hydrolyzed with PreScission Protease, isolated using a glutathione Sepharose column, and purified using Superdex-75 gel filtration chromatography. After removing endotoxin in accordance with the instructions of the Endotoxin Removal Kit (Yeasen, Shanghai, China), FIP-nha was quantified and stored at −80 °C.

### 4.3. Cell Viability Measurement

The cells were seeded in a 96-well cell culture plate at a density of 1 × 10^4^ per well and cultured in RPMI 1640 (A549 and H2347 cells) and EME (MRC-5 cell; Gibco, Grand Island, NY, USA) supplemented with 10% fetal bovine serum, penicillin (100 U/mL), and streptomycin (100 µg/mL) at 37 °C in a humidified atmosphere with 5% CO_2_. After seeding onto plates for 12 h, the cells were exposed to 0, 4, 8, 16, and 20 µg/mL FIP-nha for 24 h. Cell viability was determined using the MTT cell proliferation and cytotoxicity assay kit (Solarbio, Beijing, China) following the manufacturer’s instructions. Each experiment was repeated three times, and the viability of the control group was set to 100%.

### 4.4. In Vivo Tumor Xenograft Mouse Model

All experimental procedures involving mouse models were approved by the Laboratory Animal Ethics Committee of Beijing Institute of Genomics, Chinese Academy of Sciences under the HHS Federal Wide Assurance of Compliance Number 00014534 (7 September 2016) and IRB registration number IORG0005863 (7 September 2016).

Thirty-two 5–6-week-old male BALB/c nu/nu mice weighing 18–23 g were randomly divided into four groups. Mice were raised under specific pathogen-free conditions with a 12/12-h light/dark cycle and received chow and tap water ad libitum. Tumor xenografts were established through the subcutaneous injection of 5 × 10^6^ A549 cells (in 100 µL). After a five-day cell implantation, the four groups were intraperitoneally injected with PBS, FIP-nha (20 mg/kg body weight), FIP-nha (40 mg/kg body weight), or doxorubicin (4 mg/kg body weight) every week. Body weight and tumor size were measured every three days; tumor volume (mm^3^) was calculated based on the formula 0.5 × larger diameter × small diameter^2^.

### 4.5. iTRAQ Proteomic Analysis

A549 cells were treated with FIP-nha (8 µg/mL) or the same volume of PBS as a control for 24 h; each treatment was repeated twice. Then, whole cell proteins were collected after lysis in 4% sodium dodecyl sulfate (SDS), and digested into peptides using the filter-aided sample preparation method [43]. The tryptic peptides of FIP-nha-treated cells were labeled with the 8-plex iTRAQ reagents 113 and 114, and control peptides were labeled with 119 and 121. Four groups of peptides were mixed, fractionated by reversed phase chromatography, and detected using a Q Exactive hybrid quadrupole-orbitrap mass spectrometer (Thermo Fisher Scientific, Waltham, MA, USA) [44]. Data were processed and analyzed using Proteome Discoverer (Thermo Scientific, Waltham, MA, USA). Differentially expressed proteins were defined as having a fold change ≥1.2 or ≤0.83 and a *p*-value ≤ 0.05 for both biological repeats. Kyoto Encyclopedia of Genes and Genomes (KEGG, Kyoto, Japan) and Ingenuity Pathway Analysis (QIAGEN, Duesseldorf, Germany) were used for protein function enrichment analysis.

### 4.6. Western Blotting

A549 and H2347 cells were treated with FIP-nha (0, 4, 8, and 16 µg/mL) for 24 h. Harvested cells were lysed in 4% SDS buffer containing a protease inhibitor cocktail. Proteins were resolved using SDS-PAGE, transferred to polyvinylidene difluoride membranes (Immunobilon-P, 0.45 µm pore size; EMD Millipore, Billerica, MA, USA), and blocked with 5% non-fat milk in tris-buffered saline containing 0.1% tween-20. Then, the membranes were probed with designated primary and secondary antibodies. Immunoreactive signals were visualized using a chemiluminescent reagent kit (Merck, Darmstadt, Germany) and ImageQuant ECL (GE Healthcare, Little Chalfont, UK).

### 4.7. TEM

A549 cells were treated with FIP-nha (8 µg/mL) or the same volume of PBS as a control for 24 h. Then the cells were harvested and fixed in 2.5% glutaraldehyde and 1% osmic acid. After dehydration in a graded series of acetone, the samples were embedded in Epon812 and sectioned using Reicher-Jung Ultracut E microtome (Vienna, Austria). The ultrathin sections were stained using uranyl acetate and lead citrate, and observed using JEM1200 TEM (JEOL Ltd., Tokyo, Japan).

### 4.8. Apoptosis Assays

A549 and H2347 cells were seeded at a density of 5 × 10^5^ cells per plate in 60-mm plates. After treatment with 8 µg/mL FIP-nha and the same volume of PBS for 24 h, the cells were harvested, washed with PBS, incubated with PI and Annexin V (FITC Annexin V Apoptosis Detection Kit, BD Biosciences, San Jose, CA, USA) according to the manufacturer’s instructions, and analyzed using a BD FACSCalibur flow cytometer (San Jose, CA, USA).

### 4.9. DNA Content/Cell Cycle Analysis

A549 and H2347 cells were seeded in 60-mm plates at a concentration of 5 × 10^5^ cells/plate, and cultured in medium containing 8 µg/mL FIP-nha or PBS for 24 h. After incubation with 70% ice-cold ethanol for more than 12 h, the cells were treated successively with RNase A (50 μg/mL) at 37 °C for 15 min and PI (20 μg/mL) at 4 °C in the dark for 15 min. The DNA content was analyzed by flow cytometer (BD FACSCalibur, San Jose, CA, USA).

### 4.10. IHC Analysis

Each group contained three tumor xenografts. Tumor tissues were formalin-fixed, paraffin-enbedded, sectioned, and stained with hematoxylin and eosin. Tumor sections were deparaffinized, and the antigens were retrieved in citric acid buffer (pH 6.0) at 105 °C for 10 min. After incubation with 3% hydrogen peroxide, tumor sections were successively incubated with 10% normal goat serum for 1 h, primary antibodies against cleaved caspase 3 and Ki-67 overnight at 4 °C, and secondary antibody for 20 min at room temperature. After visualizing by incubation in 3,3-diaminobenzidine, tumors sections were counterstained with hematoxylin and digitally imaged.

### 4.11. Statistical Analysis

Data are presented as mean ± SD. The statistical significance of the differences between treatment groups and controls was determined by performing a two-tailed Student’s *t*-test using IBM SPSS software (Version 22, New York, NY, USA). *p* ≤ 0.05 were considered statistically significant.

## Figures and Tables

**Figure 1 ijms-19-03429-f001:**
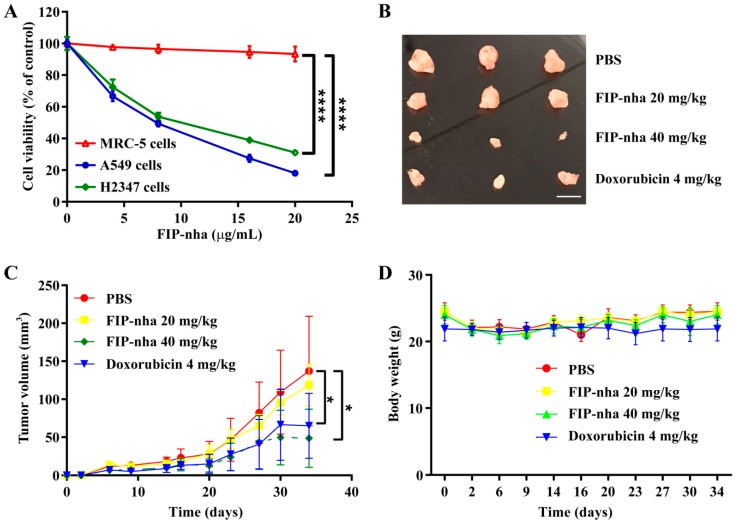
Inhibitory effects of FIP-nha on ex vivo and in vivo lung adenocarcinoma growth. (**A**) Effect of FIP-nha on MRC-5, A549 and H2347 cell growth. The cells were treated with FIP-nha (0, 4, 8, 16, or 20 µg/mL) for 24 h. Cell viability was measured by MTT assays. **** *p* ≤ 0.0001. (**B**) Inhibition of solid tumor growth in FIP-nha-treated xenograft mice of A549 cells. Images of solid tumors in negative control (PBS), positive control (doxorubicin, 4 mg/kg body weight) and experimental groups (FIP-nha, 20 and 40 mg/kg body weight) are shown. Each group contained 8 mice. Scale bar = 0.5 cm. (**C**) Effect of FIP-nha on the volume of solid tumors. * *p* ≤ 0.05 was considered significant. (**D**) Effect of FIP-nha on body weight. Quantitative data are shown as mean ± SD for tumor volume or body weight measurements.

**Figure 2 ijms-19-03429-f002:**
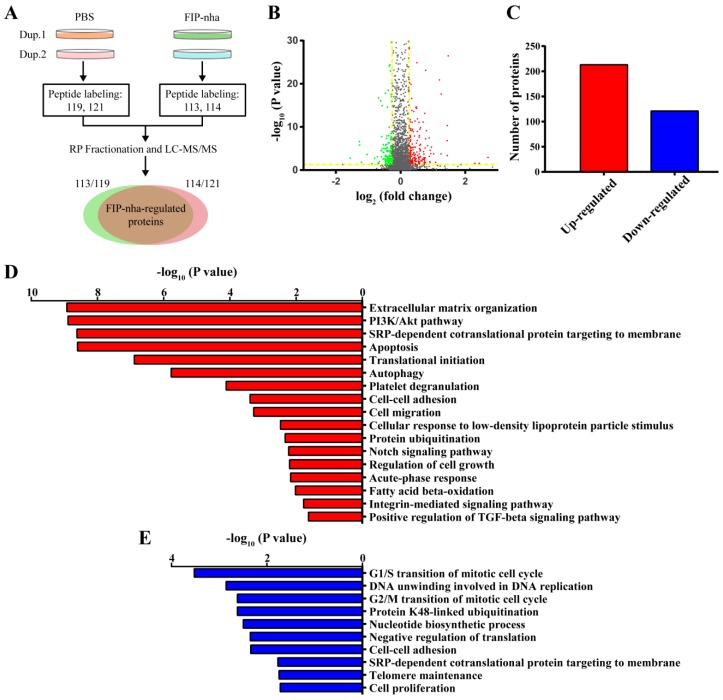
iTRAQ proteomic analysis of differential protein expression in A549 cells after FIP-nha treatment. (**A**) Experimental flow diagram of proteomic analysis. Dup: biological duplicate; RP: reversed phase chromatography. (**B**) Volcano plot of identified proteins. The proteins were indicated in color as follows: green, fold change ≤ 0.83, *p* ≤ 0.05; red, fold change ≥ 1.2, *p* ≤ 0.05; gray, 0.83 < fold change or *p* > 0.05; yellow lines, threshold lines of fold change and *p* value. (**C**) The number of differentially expressed proteins. (**D**) Functional category enrichment of the upregulated proteins. The *y*-axis indicates significantly enriched function terms and the *x*-axis shows the enrichment *p* value. (**E**) Functional category enrichment of downregulated proteins.

**Figure 3 ijms-19-03429-f003:**
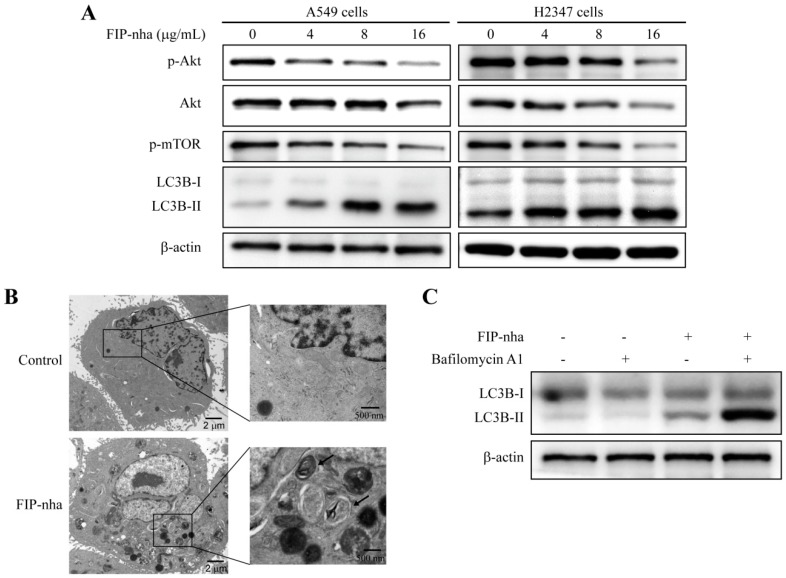
FIP-nha negatively regulates the PI3K/Akt signaling pathway and induces autophagy in lung adenocarcinoma cells by downregulating p-mTOR. (**A**) A549 and H2347 cells were treated with 0, 4, 8, and 16 µg/mL FIP-nha for 24 h. The protein expression levels of p-AKT, AKT, p-mTOR, and LC3B-I/II were examined by western blotting. (**B**) Ultrastructures of autophagosome in A549 cells treated FIP-nha (8 µg/mL) for 24 h. Arrows indicate autophagosome vesicles contained double membranes and damaged organelles. (**C**) A549 cells were treated with FIP-nha (8 µg/mL) for 24 h and then bafilomycin A1 (50 nM) for another 4 h. The protein levels of LC3B-I/II were examined by western blotting.

**Figure 4 ijms-19-03429-f004:**
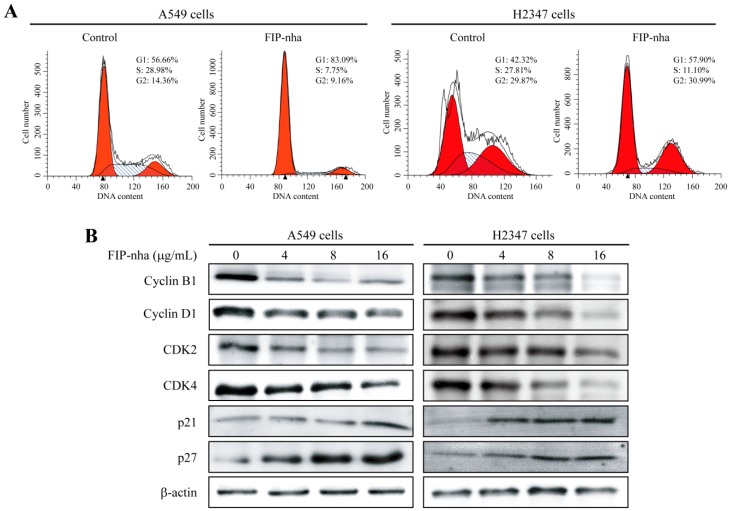
FIP-nha induces G1/S and G2/M cell cycle arrest in lung adenocarcinoma cells. (**A**) A549 and H2347 cells were treated with 8 µg/mL FIP-nha for 24 h and cell cycle distribution was determined by flow cytometry. Untreated cells were used as control. (**B**) Western blotting analysis of cyclin B1, cyclin D1, CDK2, CDK4, p21, and p27 protein expression in A549 and H2347 cells after exposure to FIP-nha at 0, 4, 8 and 16 µg/mL for 24 h.

**Figure 5 ijms-19-03429-f005:**
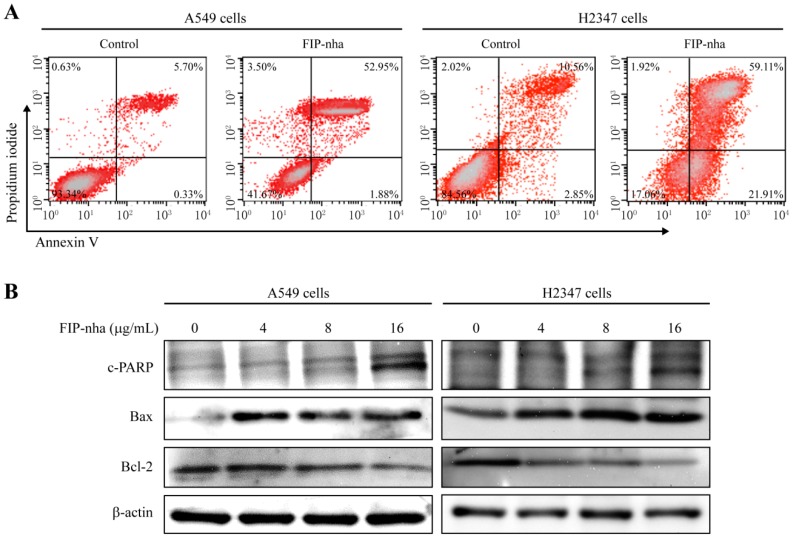
FIP-nha induces apoptosis in A549 and H2347 cells. (**A**) A549 and H2347 cells were incubated with 8 µg/mL FIP-nha for 24 h. Untreated cells were used as control. After labeling with Annexin V-FITC and propidium iodide (PI), apoptotic cells were detected by flow cytometry. (**B**) The expressions of apoptosis-related proteins c-PARP, Bax, and Bcl-2 were assessed by western blotting.

**Figure 6 ijms-19-03429-f006:**
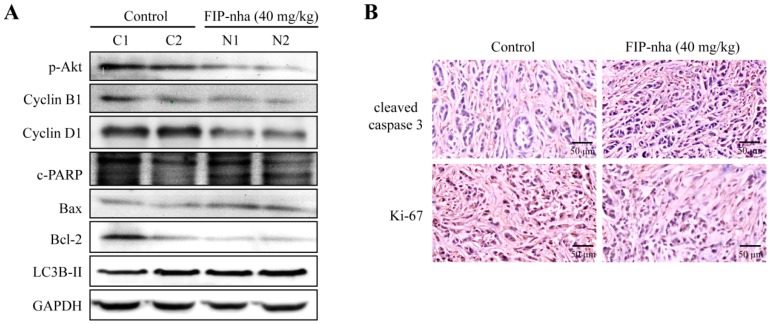
(**A**) The levels of p-Akt, cyclin B1, cyclin D1, c-PARP, Bax, Bcl-2, and LC3B-II in xenograft tumors were determined by western blotting. The experimental group (N1 and N2) was treated with 40 mg/kg FIP-nha, and the control group (C1 and C2) was treated with the same volume of PBS. (**B**) IHC of cleaved caspase 3 and Ki-67 in xenograft tumors. Each group contained three mouse tumors.

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
