# Peer review of "Fungal Immunomodulatory Protein from Nectria haematococca Suppresses Growth of Human Lung Adenocarcinoma by Inhibiting the PI3K/Akt Pathway"

_ijms, 2018, doi:10.3390/ijms19113429_

Round 1
Reviewer 1 Report
Xie et al. present interesting results on the effects of fungal immunomodulatory protein from Nectria Haematococca (FIP-nha) on A549 human lung cancer cells in vitro and in vivo. The mechanisms underlying the observed anti-cancer effects were elucidated by comparative quantitative proteomics. FIP-nha treatment suppressed PI3K/Akt signaling and induced cell cycle arrest, apoptosis and autophagy. Overall, the study is well designed but there are several suggestions to improve the quality of the manuscript:
1) In figure 1A, it would be beneficial if the effects of FIP-nha were shown on more than one NSCLC cell line. This helps to illustrate the fact that FIP-nha has broad effects on NSCLC cells and is not specific to a particular cell line.
2) Figure 6 could be improved further if the effects of FIP-nha on the proliferation and apoptosis of A549 cells in vivo were demonstrated through immuno-histochemistry (i.e. Cleaved caspase 3, Ki67 IHCs).
3) For figure 3B, autophagy flux assays (as described in guidelines for autophagy assays : PMID: 26799652) are required to provide a more accurate definition of autophagy induction. (e.g. blot for LC3B +/- BafilomycinA1).
Reviewer 2 Report
In this study the authors describe the effect of fungal protein FIP-nha in the tumor growth in A549 lung cancer cell line. Additionally showed a proteomics analysis where they found that FIP-nha negatively regulates PI3K/Akt signaling and induces cell cycle arrest, autophagy, and apoptosis. Demonstrated that FIP-nha suppresses Akt phosphorylation, up-regulation of p21 and p27 and down-regulation of cyclin B1, cyclin D1, CDK2, and CDK4 expression, resulting in cell cycle arrest. Overall reveal the potential anti-tumor role of FIP-nha and provided a potential further application of this fungal proteins to cancer treatments.
1) At least I suggest that the results of figures 1, 3, 4, 5 and 6 be replicated in other lung adenocarcinoma cell lines and in the text I suggest that the authors be cautious in interpreting the data referred to this histological subtype (Lung adenocarcinoma). The same for the title of this article.
2) The results of figure 2, specifically figure 2D, show category enrichment of the up-regulation proteins. Where one of the main upregulated pathways are PI3K/AKT and Autophagy under the treatment with FIP-nha, suggesting that FIP-nha induces greater activity of these pathways. However, the authors interpret this as an inhibitory effect of these pathways and the subsequent experiments of Figs. 3 and 6 concentrate on demonstrating this inhibitory effect. Could the authors clarify these results and their interpretation?
3) In figure 5b, I suggest moving the order of the first c-PARP, BAX and then Bcl-2 figures.
4) In figure 6, the western blot of c-PARP is unclear and no differences are observed
5) In the discussion a series of supplementary results are mentioned that I suggest to take them out of the discussion and move them to the results and leave the discussion only to discuss and highlight the relevant results.
Author response to report 1:
1)A: Thanks for your great suggestion. We have evaluated the effect of FIP-nha on another lung adenocarcinoma cell, NCI-H2347, and have added the results in the revised manuscript.
2)A: Thanks for your suggestion. We have added immunohistochemistry analysis of cleaved caspase 3 and Ki-67 in tumor xenografts in the revised manuscript.
3)A: Thanks for your advice. Autophagic vacuoles detection using transmission electron microscope and western blotting for LC3B +/-Bafilomycin A1 have been added in the revised manuscript.
Author response to report 2:
1)A: Thanks for your great suggestions. We repeated these experiments in another lung adenocarcinoma cell NCI-H2347, and got similar results, which have been revised in the manuscript. We are sorry for the incautious interpreting, and we have revised “NSCLC” to “lung adenocarcinoma” in the text and the title.
2)A: Thanks for your questions. In the functional category enrichment analysis, the proteins are up-regulated, which does not mean that the regarding pathways or biological processes are activated. When the identified proteins play negative roles or are inhibitors of some key factors in the pathways, which may lead to the inhibition of these pathways. We have checked the identified proteins enriched in PI3K/Akt signaling pathway, and found most of the proteins, such as phosphatase and tensin homolog (PTEN) and thrombospondin-1 (THBS1), negatively regulate this pathway. In addition, activation PI3K phosphorylates and activates Akt, which subsequently affects its downstream factors, such as phosphorylating mTOR. In this pathway, alteration of protein modification (phosphorylation) is significant. In our study, we used iTRAQ proteomic analysis to quantify the protein expression level, but not the modification (phosphorylation) alteration, which may be the reason why down-regulated proteins were not enriched in the PI3K/Akt signaling pathway.
3)A: Thanks for your suggestion, and we have revised in the manuscript.
4)A: Thanks. We repeated and optimized this experiment, and corresponding revise have been made in the manuscript.
5)A: Thanks for your suggestion. We have made corresponding revise in the manuscript
Round 2
Reviewer 1 Report
The authors have made significant changes, based on the previous comments provided. The quality of the manuscript has been improved and is worthy of publication.
Reviewer 2 Report
Congrats!!!, you did a good job of properly addressing the comments and working hard to respond quickly.
Thanks